# Ergodic Subspace Analysis

**DOI:** 10.3390/jintelligence8010003

**Published:** 2020-01-06

**Authors:** Timo von Oertzen, Florian Schmiedek, Manuel C. Voelkle

**Affiliations:** 1Department of Psychology, University of the Federal Forces Munich, 85579 Neubiberg, Germany; 2Max Planck Institute for Human Development, 14195 Berlin, Germany; 3DIPF | Leibniz Institute for Research and Information in Education, 60323 Frankfurt am Main, Germany; schmiedek@dipf.de; 4Department of Psychology, Humboldt-Universität Berlin, 10117 Berlin, Germany; manuel.voelkle@hu-berlin.de

**Keywords:** ergodicity, dimension reduction, ergodic subspace analysis, cognition

## Abstract

Properties of psychological variables at the mean or variance level can differ between persons and within persons across multiple time points. For example, cross-sectional findings between persons of different ages do not necessarily reflect the development of a single person over time. Recently, there has been an increased interest in the difference between covariance structures, expressed by covariance matrices, that evolve between persons and within a single person over multiple time points. If these structures are identical at the population level, the structure is called ergodic. However, recent data confirms that ergodicity is not generally given, particularly not for cognitive variables. For example, the *g* factor that is dominant for cognitive abilities between persons seems to explain far less variance when concentrating on a single person’s data. However, other subdimensions of cognitive abilities seem to appear both between and within persons; that is, there seems to be a lower-dimensional subspace of cognitive abilities in which cognitive abilities are in fact ergodic. In this article, we present ergodic subspace analysis (ESA), a mathematical method to identify, for a given set of variables, which subspace is most important within persons, which is most important between person, and which is ergodic. Similar to the common spatial patterns method, the ESA method first whitens a joint distribution from both the between and the within variance structure and then performs a principle component analysis (PCA) on the between distribution, which then automatically acts as an inverse PCA on the within distribution. The difference of the eigenvalues allows a separation of the rotated dimensions into the three subspaces corresponding to within, between, and ergodic substructures. We apply the method to simulated data and to data from the COGITO study to exemplify its usage.

## 1. Introduction

Ergodicity (see [26] for the first use of the term; for a more general discussion in the behavioral sciences, see [18]) in the behavioral sciences describes how the interrelation seen between some variables between persons is the same within a person. For example, persons who are taller are typically also heavier; that is, height and weight are positively correlated between persons. At the same time, if we observe the growth of height in a child, we also expect his or her weight to increase, so that height and weight are also positively correlated within the same person. If these two correlations are identical, we say that height and weight are ergodic.[note 1]

The assumption of ergodicity allows conclusions from between-person data, which typically is more efficiently available in applied research, to be applied to within-person situations. In our example, if we assume ergodicity and learn that a specific child has grown by some factor, we can reasonably assume that the child will also be heavier, even if we lack any within-person data about this specific child.

However, ergodicity is by far not present in all situations. The main theoretical reason that can lead to a reduction of ergodicity is a situation in which two aspects of performance depend on the same resource. Then, it can be assumed that the variables are positively related between persons because of inter-individually different availability of this resource. At the same time, the two variables can be assumed to be negatively correlated within the same person, who balances her available resources between the two tasks. For example, consider the task of writing a text on a typewriter quickly and accurately. Since both require similar cognitive resources, it can be expected that a person who is a very fast typist will at the same time also be more accurate. However, if a specific person is pressed to type faster than her usual long-term mean, she may be expected to produce more errors. Conversely, if she pays particular attention to reducing the probability of an error, this will probably come at the cost of lower speed. In the same vein, the quantity and quality of scientific publications of researchers can be expected to be positively correlated between persons; that is, a researcher who is able to publish more is in expectation also more likely to find outstanding new results. However, within a person, the process of disseminating her work will likely cost resources she cannot invest in creating new research, thus inducing a negative correlation of publication quantity and quality within the same person over time (see also [11]).

[3] describe similar effects empirically in personality psychology. They found that the well-known structure of personality between persons is not necessarily the same within a single person over time. [6] analogously observed that the structure of affect-related variables is different between persons from the structure of the same variables within the same person observed over 100 days. In particular, negative and positive effect within the same person cannot be conceived as two orthogonal dimensions, as is the case between persons. Given the same participants, [29] showed that even the cognitive structure of one person over time cannot be inferred from the between-person structure of cognitive ability; while between persons, cognitive abilities form one positive manifold, this is not true for specific persons over time, where considerable divergences from the *g*-factor model can be observed. The difference can be assessed by computing the Kullback–Leibler divergence between a *g*-factor model, fit to the within data of a single person over the 100 days, to the actual covariance matrix, which essentially is the $\chi^{2}$ of the model; this value is larger for some participants compared to the between-person model. Even between persons, the role of the *g* factor as a latent variable is easily understood and often debated in different contexts (see [5] for a general discussion and [4] for a related debate in this area).

A lack of ergodicity limits our ability to draw conclusions from between-person data. However, since this data is much more readily available than longitudinal studies, we are tempted to draw such conclusions anyway, resulting in an ecological fallacy. This may lead to severely wrong conclusions when trying to predict an outcome from a single person’s history. [8] describe this and a number of further threats that human subject research faces when ergodicity is not given (but see [1]).

In general, the problem of insufficient ergodicity can always be overcome by using longitudinal panel data. [12] describe a number of ways in which time series analysis can be used to create models of an individual, and many others have suggested methods with the same aim (for an example, see [32]).

However, as is the case with most properties in psychological science, ergodicity is typically not either “present” or “not present” but should be seen as a quantitative property that exists for some part of the data, but not for all (cf. [30]). Methods to identify the degree of ergodicity in a given set of variables are scarce, and in particular regarding the parts of the data in which ergodicity is present and in which it is not. This problem can be seen analogously to a dimension-reduction problem: Even though a dataset is initially high-dimensional, sometimes only a small number of dimensions are necessary to describe what is considered important for some analyses. The best-known example is a principal component analysis (PCA, [25]), in which a subspace of the variable space is identified that is lower-dimensional but still explains most of the variance in the dataset. Other dimension-reduction methods identify subspaces with different aims. For example, slow feature analysis (SFA, [33]) tries to identify dimensions in the data that change the least over time, assuming that these dimensions are most informative for properties in a time series that is constant over long periods of time and only changes infrequently, but rapidly. Common spatial patterns (CSP, [15]) focuses on the subspace in which two groups differ most strongly, with the main aim of allowing for better classification between the two groups.

In this article, we adapt the CSP by [15], providing an alternative method called ergodic subspace analysis (ESA) to identify three subspaces of the variables: one in which the between structure and the within structure are mostly ergodic, one in which the between structure is dominant, and a third one in which the within structure is dominant. The factors of the first subspace can be seen as a combination of variables that are valid both between persons and within persons. That is, if a person scores highly on a variable that is important for one of these factors, this person can generally be expected to score high on all other variables with high loadings for this factor as well, and low on the others. At the same time, if a person loads unusually high on this variable on a specific day compared to her mean score over time, then all important variables of this factor can be expected to be higher on this day compared to their respective means. For the second subspace, the same logic holds between persons; that is, a generally high value on one variable that is important for such a factor implies high values on other variables with high loadings on this factor. However, if that variable is high at a given time for a participant, her score on the other variables loading high on the factor will not necessarily be higher than usual. The opposite holds for factors on the third group: Here, predictions from variables with high loadings can be made over time, but not between different participants.

## 2. Definition of Ergodicity

If we assume that *K* variables have been measured over t=1,…,T time points on i=1,…,N persons, then the *K* variables at every pair (i,t) have a specific distribution. In the following, we will concentrate only on the covariance matrix $\Sigma_{i,t}$ for the *i*th person at the *t*th time point. A strict definition of ergodicity ([30]) requires that all these distributions are identical. In particular, for the covariance matrices, we get that:(1)$$\forall \left(i=1,\ldots ,N,t=1,\ldots ,T\right):\Sigma_{i,t}=\Sigma$$
for a fixed covariance matrix Σ. This requirement can be seen as the union of stationarity for all participants, defined as:(2)$$\forall i=1,\ldots ,N:\Sigma_{i,1}=\ldots =\Sigma_{i,T}=\Sigma_{i}$$
for covariance matrices $\Sigma_{i}$ and homogeneity for all time points defined as:(3)$$\forall t=1,\ldots ,T:\Sigma_{1,t}=\ldots =\Sigma_{N,t}=\Sigma_{t}$$
for covariance matrices $\Sigma_{t}$. A weaker definition of ergodicity (weak ergodicity) requires only that the marginal distributions for all participants and for all time points be identical. The marginal distribution of any participant is the distribution of this participant assuming *T* identical independent draws from a single distribution, and analogously for time points. Computationally, the marginal covariance matrix of participant *i* and time point *t*, respectively, are given by the average of the corresponding covariance matrices:(4)$$\begin{array}{ccc} \Sigma_{i,t=1\ldots T} & = & \frac{1}{N}\sum_{t=1}^{T}\Sigma_{i,t} \end{array}$$(5)$$\begin{array}{ccc} \Sigma_{i=1\ldots N,t} & = & \frac{1}{T}\sum_{i=1}^{N}\Sigma_{i,t} \end{array}$$

The weak stationarity definition then requires that:(6)$$\begin{array}{cc} \forall i=1,\ldots ,N: & \Sigma_{i,t=1,\ldots ,T}=\Sigma \end{array}$$(7)$$\begin{array}{cc} \forall t=1,\ldots ,T: & \Sigma_{i=1,\ldots ,N,t}=\Sigma \end{array}$$
again for a fixed covariance matrix Σ.

Note that although some necessary conditions for strict ergodicity can be tested, it is not possible to design a sufficient and necessary condition that can be tested since no repeated instances of random draws from a pair (i,j) can be sampled. This would require that we can observe the same person at the same point in time multiple times. Hence, no covariance matrix for a specific person at a specific time point can be estimated, and since every exception, e.g., a single time point for a single person with a different covariance matrix, strictly speaking destroys ergodicity, the strict version of ergodicity cannot be tested. However, with simple monotonicity assumptions, we can come close to testing ergodicity (see [30]). In fact, in most if not all practical cases, we expect that any violation of strict ergodicity will also violate the conditions on the marginals for weak ergodicity, even though, theoretically, weak ergodicity does not necessarily imply strict ergodicity. ESA, described in the current article, identifies subspaces between any two covariance matrices and can thus be used for both definitions, provided that approximations to either the marginal or the point covariance matrices are available.

Note that the time and participant dimensions are not symmetrical to each other, as may seem at first glance. The reason for this lies in the definition of the two; we assume that (besides potential global means, which are not important for this consideration) every participant has a mean value, which is an instance of the between distribution, and then has a day-to-day fluctuation, which is, each day, an instance of the within distribution. As closer inspection shows, this means that in the generative process—regardless of whether it is artificially done in a simulation or in reality—there are *N* samples drawn from the between covariance matrix but N×T samples drawn from the within covariance matrix. In effect, a higher number of participants helps to find ergodicity, or a lack thereof, more than a higher number of time points.

### Conditional Equivalence

[30] introduced the concept of conditional equivalence. A set of variables is conditionally equivalent if ergodicity[note 2] is given when controlling for covariates (potentially including indirectly measured parameters, e.g., an autoregression parameter) that are either constant over time or over persons. For example, a time-specific covariate could be seasonal effects, a person-specific covariate could be gender.

In the example comparing the dependency of speed and accuracy in typewriting, the general typewriting ability could be such a covariate, which can be assumed to be constant within a person over different time points but varies between persons. If we control for this ability, we would no longer expect a positive covariance between speed and accuracy between persons, and thus would have approached an ergodic, or at least more ergodic, situation. If the data are completely ergodic when controlling for ability, we would say that the situation is conditionally ergodic. [30] suggest some methods for estimating the conditional covariance matrices can be estimated with Kalman filters (for a more didactical introduction, see [31]).

Conditional equivalence can be perceived as a confirmatory approach that models specific external covariates such that subspaces identified by the same value of this covariate are ergodic. ESA searches these subspaces in an exploratory fashion, so ESA can be perceived as the exploratory sibling of conditional equivalence.

## 3. Ergodic Subspace Analysis

We will now introduce ergodic subspace analysis. The method requires two covariance matrices $\Sigma_{BP}$ and $\Sigma_{WP}$ as input, where $\Sigma_{BP}$ is a between-participant (BP) covariance matrix and $\Sigma_{WP}$ is a within-participant (WP) covariance matrix. If a dataset with *N* participants measured over *T* time points is given, the most straightforward candidates for these matrices are the average of between covariance matrices and the average of all within covariance matrices. The method will eventually output three subspaces, defined by orthogonal vectors, that represent a between space (in which the between covariance matrix shows dominant variance), a within space (in which the within covariance matrix shows dominant variance), and an approximately ergodic subspace.

For a real dataset, there are different possibilities for obtaining these two matrices. The option that needs the least data is to collect one dataset of *N* persons cross-sectionally and to follow a single person over *T* time points. If we assume that the marginal distributions with respect to *N* and *T*, respectively, are sufficiently identical, this is a sufficient approach for applying ESA to find ergodic and non-ergodic subdimensions. However, this assumption cannot be tested from such data. It has been shown that at least in the cognitive domain, while the difference in the between matrices over time may not be too strong, the difference between persons can be substantial ([29]). Nevertheless, ESA can still be applied even without this assumption, for example, to find ergodic or non-ergodic subspaces between this specific person and the between situation.

An alternative approach is to follow multiple persons in the dataset longitudinally, and ideally all participants, to get a full N×T set of data. The within covariance matrix can be obtained by estimating the covariance for every single participant and then averaging those matrices. If we assume that the between means on the different time points are sufficiently similar, the best way to obtain the between covariance matrix is to take the average value for every participant and take their covariance matrix. Otherwise, we can estimate the covariance matrix on all N×T data points and then subtract the estimated within covariance matrix from the estimate.

Once we have obtained an estimate for the within-person and between-person covariance matrix, we perform the ESA in four steps. The four steps are as follows:

In Step 1, we combine both matrices for an overall covariance matrix Σ:(8)$$\Sigma =\frac{\Sigma_{WP}+\Sigma_{BP}}{2}$$

In Step 2, we perform a whitening on Σ (see, for example, [24]). For this, we first compute an eigenvector decomposition $Q_{1}$ of Σ,
(9)$$Q_{1}\Sigma Q_{1}^{T}=D_{1}$$
with $D_{1}$ being a diagonal matrix with the eigenvalues of Σ on the diagonal, which by default we assume to be ordered by size. We then divide each row and column of $Q_{1}$ by the square root of the eigenvalue,
(10)$$Q_{white}=\mathrm{diag}\left(\frac{1}{\sqrt{\lambda_{1}}},\ldots ,\frac{1}{\sqrt{\lambda_{K}}}\right)Q_{1}$$
with this transformation, Σ is orthogonally transformed to the identity matrix:(11)$$Q_{white}\Sigma Q_{white}^{T}=I$$
the rationale behind the whitening is to remove all normal information, that is, all information from the first and second moments, from the common between covariance and within covariance matrix. Note that the transformation is linear on the addition of Step 1, so that we have:(12)$$\begin{array}{ccc} Q_{white}\frac{1}{2}\Sigma_{BP}Q_{white}^{T}+Q_{white}\frac{1}{2}\Sigma_{WP}Q_{white}^{T} & = & Q_{white}\Sigma Q_{white}^{T} \end{array}$$(13)$$\begin{array}{cc} = & I \end{array}$$

In Step 3, we compute an eigenvector decomposition $Q_{2}$ on the transformed between covariance matrix such that
(14)$$Q_{2}\left(Q_{white}\frac{1}{2}\Sigma_{BP}Q_{white}^{T}\right)Q_{2}^{T}=D_{BP}$$
where, again, $D_{BP}$ is a diagonal matrix with the eigenvalues $\lambda_{1\;BP},\ldots ,\lambda_{K\;BP}$ of the transformed between covariance matrix, as usual assumed to be ordered from large to small eigenvalues. We now define the resulting total transformation as the ESA transformation $Q_{ESA}$:(15)$$Q_{ESA}=Q_{2}Q_{white}$$

Note that the ESA transformation on the within covariance matrix $\Sigma_{WP}$ also yields a diagonal matrix $D_{WP}$ with entries $1-\lambda_{k\;BP}$:(16)$$\begin{array}{ccc} Q_{ESA}\frac{1}{2}Q_{WP}Q_{ESA}^{T} & = & Q_{2}\left(Q_{white}\frac{1}{2}\Sigma_{WP}Q_{white}^{T}\right)Q_{2}^{T} \end{array}$$(17)$$\begin{array}{cc} = & Q_{2}\left(I-Q_{white}\frac{1}{2}\Sigma_{BP}Q_{white}^{T}\right)Q_{2}^{T} \end{array}$$(18)$$\begin{array}{cc} = & Q_{2}Q_{2}^{T}-Q_{ESA}\frac{1}{2}\Sigma_{BP}Q_{ESA}^{T} \end{array}$$(19)$$\begin{array}{cc} = & I-D_{BP} \end{array}$$

Since all covariance matrices are positive (semi)definite, both $\lambda_{k\;BP}$ and $\lambda_{k\;WP}=1-\lambda_{k\;BP}$ must be larger than or equal to zero, and since they sum up to one, they must be in the interval [0,1]. The entries of $D_{WP}$ are in particular sorted from the smallest to the largest entry.

The difference $erg_{k}=\lambda_{k\;BP}-\lambda_{k\;WP}$ of an eigenvalue pair lies between −1 and +1 and is a quantitative measure for how ergodic this component is: If $erg_{k}$ is zero, then both the within and the between covariance matrix have the same variance. The higher the value is, the more variance can be found between persons relative to the within-person variance for this factor. Negative values, on the other hand, suggest that there is more variance within a person than between persons. A theoretical value of 1 would indicate that the component appears only in the between-person data, and −1 that the component appears only in the within-person data.

In Step 4, we fix a cutoff *c* to indicate which absolute value of $erg_{k}$ we want to accept as sufficiently ergodic. A straightforward value seems to be c=0.1; that is, any factor with an ergodicity value between −0.1 and 0.1 would be considered ergodic. This means that the variances differ at most by 10 percent of the total variance in this direction. However, any other cutoff value is possible depending on how conservative the classification of “ergodic” is chosen to be. We would like to caution readers against assuming that any fixed value for the cutoff is optimal in all situations; although 10 percent of the total variance might be a good choice in many situations, the value should be higher, e.g., c=0.2, if it is important to be conservative in identifying between or within factors, and should be lower, e.g., c=0.05, when one aims to be conservative in determining factors to be ergodic.

The rows of $Q_{ESA}$ with $|erg_{k}|\leq c$ span a subspace that is ergodic by this definition, while the rows with $erg_{k}>c$ and $erg_{k}<-c$ span subspaces with higher variance between or within persons, respectively. Formally, the method outputs the three subspaces:(20)$$\begin{array}{ccc} V_{BP} & = & \langle Q_{k\;ESA}|erg_{k}>c\rangle \end{array}$$(21)$$\begin{array}{ccc} V_{\mathrm{ergodic}} & = & \langle Q_{k\;ESA}|-c\leq erg_{k}\leq c\rangle \end{array}$$(22)$$\begin{array}{ccc} V_{WP} & = & \langle Q_{k\;ESA}|erg_{k}<-c\rangle \end{array}$$

These three subspaces together fill the whole variable space and have the desired property. In particular, the middle space $V_{\mathrm{ergodic}}$ is a subspace of the variables that is quantitatively close to an ergodic space.

An implementation of the method can be found in Appendix A. It also includes code for the following two examples.

## 4. Performance in Artificial Situations

In this section, we demonstrate how ESA works on artificial examples. We first manually calculate a very simple example to demonstrate the mathematics and then perform a number of simulations using the backend of the graphical structural equation modeling (SEM) program Onyx ([23]).

### 4.1. Computation Examples

#### 4.1.1. Manual Computational Example

We first consider a very simple situation that can be followed without the help of a computer, based on our initial example from the introduction. Assume we have measured N=40 participants on T=40 different sheets of paper on which they typed on a typewriter, sampling their accuracy (e.g., ratio of correct key hits) and their speed (e.g., key hits per minute). We assume that every participant has a general ability that creates a between-participant correlation of 0.5 of accuracy and speed. At the same time, we assume that every single participant either allocates her resources to speed or accuracy, creating a −0.5 correlation within each participant. For reasons of simplicity, we assume that variances are one. The total covariance matrix in this example is then:(23)$$\Sigma =\frac{1}{2}\left(\begin{array}{cc} 1 & 0.5\\ 0.5 & 1 \end{array}\right)+\frac{1}{2}\left(\begin{array}{cc} 1 & -0.5\\ -0.5 & 1 \end{array}\right)=\left(\begin{array}{cc} 1 & 0\\ 0 & 1 \end{array}\right)$$

Since this matrix is the identity matrix, the total distribution is already white. The diagonalization matrix $Q_{1}$ as well as the diagonal matrix in Step 1 are, therefore, also the identity matrix, which means that $Q_{white}=I$. We now find $Q_{2}$ as the diagonalization of the transformed between-person covariance matrix $Q_{white}\frac{1}{2}\Sigma_{BP}Q_{white}^{T}$, which in this simple example is:(24)$$\frac{1}{2}\left(\begin{array}{cc} 1 & 0.5\\ 0.5 & 1 \end{array}\right)$$

This matrix has the eigenvectors $\left(\sqrt{0.5},\sqrt{0.5}\right)$ and $\left(\sqrt{0.5},-\sqrt{0.5}\right)$, for the eigenvalues 0.75 and 0.25. The total ESA transformation matrix is therefore:(25)$$Q_{ESA}=Q_{2}Q_{white}=\left(\begin{array}{cc} \sqrt{0.5} & \sqrt{0.5}\\ \sqrt{0.5} & -\sqrt{0.5} \end{array}\right)$$
the corresponding eigenvalues of the within-person data are $\lambda_{1\;WP}=1-\lambda_{1\;BP}=0.25$ and 0.75 analogously, which gives $erg_{1}=0.5$ and $erg_{2}=-0.5$. With any cutoff below 0.5, we would hence find no ergodic subspace (which is plausible given the example), a between-dominated subspace that is spanned by the vector (1,1), and a within-dominated subspace spanned by (1,−1), which is exactly as we constructed the data.

#### 4.1.2. Larger-Scale Computational Example

In this section, we consider a larger example that we cannot compute manually. Assume we measure participants in four sports disciplines, repeating the measurement for some days. The data is generated as a combination of four orthogonal factors:(26)$$\begin{array}{ccc} v_{1} & = & \left(1,1,1,1\right) \end{array}$$(27)$$\begin{array}{ccc} v_{2} & = & \left(3,-1,-1,-1\right) \end{array}$$(28)$$\begin{array}{ccc} v_{3} & = & \left(0,1,-1,0\right) \end{array}$$(29)$$\begin{array}{ccc} v_{4} & = & \left(0,1,1,-2\right) \end{array}$$

The factors were normalized before being used; the integer representation is used here only to assist in understanding the conceptual influence of the factors.

We assume the first factor, which in the example describes something like general fitness, to be strong between but weak within participants. The second factor describes how strongly participants concentrate on the first discipline, which, because they become tired, reduces their ability to perform well on the remaining three disciplines. This factor is generated to be particularly strong within persons. The third factor, which describes a trade-off between the second and third disciplines, is assumed to be ergodic. The remaining factor is weaker but also ergodic. So ideally, the first two factors should be sorted into between and within spaces, while the remaining two should both span the ergodic subspace. This is represented in the loadings for between and within as follows:(30)$$\begin{array}{ccc} f_{BP} & = & \left(10,x,5,1\right) \end{array}$$(31)$$\begin{array}{ccc} f_{WP} & = & \left(x,10,5,1\right) \end{array}$$

Both covariance matrices $\Sigma_{BP}$ and $\Sigma_{WP}$ were then generated from these factors. The *x* was different between simulation conditions, being first set to zero (for the extreme case) and then increased in steps of 1 up to 10 to show what happens when the dataset overall becomes more and more ergodic.

For data generated from these matrices, the expected outcome of an ESA can be computed directly for every *c*. For the extreme case x=0, the ESA in expectation returns the factors (in the rows of the matrix; note that the signs can be switched and the middle two rows can be mixed since they share the same ergodicity value for x=0):(32)$$Q_{ESA}=\left(\begin{array}{cccc} 0.224 & 0.224 & 0.224 & 0.224\\ -0.000 & -0.408 & -0.408 & 0.816\\ 0.000 & 0.316 & -0.316 & 0.000\\ -0.387 & 0.129 & 0.129 & 0.129 \end{array}\right)$$
with the ergodicity values for the four factors (in the order returned through the method, that is, sorted from largest to smallest) being:(33)erg=(1,0,0,−1)
which exactly reflects the data-generation situation.

Figure 1 shows the first estimated ergodicity value (that is, $erg_{1}$) for different values of *c*. For all these values, $erg_{4}=-erg_{1}$ and $erg_{2}=erg_{3}=0$. As can be seen, the expected estimated ergodicity follows exactly the square of 1−x; that is, the more ergodic the two first generating factors become, the lower is their ergodicity value, until it reaches the threshold of 0.1 for x=9 (when nine-tenths of the variance in the factors occurs both between and within participants) and finally reaches zero for a fully ergodic situation.

### 4.2. Simulations

#### 4.2.1. Simulation 1

Using the sports example from the last section, we simulated data in 5000 trials and performed the ESA on the data itself to investigate the standard error under different conditions. Data was generated by first choosing a random mean value for every participant from a normal distribution with the between covariance matrix. Then, for this person a value was randomly chosen from a normal distribution for every time point using the within covariance matrix and was added to the participant mean. The mean for both samplings was set to zero, so that the overall mean over all persons was again zero.

We first generated data for N=100 and T=100 observations with *x* again varying between 0 and 10. The factors were reconstructed perfectly for small *x*, as expected from the computation above. For larger *x* (when $v_{1}$ and $v_{2}$ also become ergodic), the reconstruction was more influenced by the sampling error. We measured this by the cosine of the angle between $v_{1}$ and the first row of $Q_{ESA}$, which is one for a perfectly reconstructed first factor and 0 if both vectors are orthogonal. We also measured the ergodicity value of the first factor in the ESA. For this value, we controlled for the effect of the incomplete reconstruction of $v_{1}$ by measuring the actual variance in the direction of $v_{1}$.

Results are shown in Table 1. For the perfectly separated case (first row), the reconstruction is perfect. As ergodicity grows, the first three vectors become increasingly similar, and the first factor of the ESA finds any combination of them, which reduces cos(α). The values for $erg_{i}$ almost perfectly correspond to the values theoretically predicted above. As can be seen, the standard error of the ergodicity value reduces with larger ergodicity, which means that researchers are more likely to find values close to zero if the data are fully ergodic than to find values close to one if one factor is only present between participants. Note that the errors are not normally distributed for these high values since 1.0 is an upper ceiling for the ergodicity value.

#### 4.2.2. Simulation 2

To investigate the effect of sample size and time on the standard deviation, we repeated the same simulation with varying *N* and *T* from 25 to 250 in steps of 25, and keeping *x* constant at a medium ergodicity of five out of ten (the middle row in Table 1). Since the reconstruction rate of the first factor for this *x* was (with cos(α)=0.938) still fairly high and did not vary substantially for different sample sizes, and likewise the mean value for $erg_{1}$ controlling for the remaining small reconstruction errors did not vary under the different conditions, we only report the standard deviation of the first ergodicity value. Again, we performed 5000 trials in each condition.

Results are shown in Table 2 and Figure 2. The standard error of the ergodicity measure ranges from 0.2 for the 25×25 observations to 0.06 for 250×250 observations. The point at N=100 and T=100 is the same as the point with c=5 in Table 1; the small difference is due to the simulation error. As expected, the two dimensions are not symmetrical with respect to *N* and *T*, with *N* being considerably more important and only a minimal tendency for *T* to show more precision when having more time points. The standard error mostly follows the $\frac{1}{\sqrt{N}}$ law we would expect for sample size (that is, if we quadruple sample size, the measurement error is reduced to half its original value).

#### 4.2.3. Simulation 3

The *K* ergodicity values can be plotted in a scree plot, similar to eigenvalues in a PCA, to assist the researcher in finding appropriate cutoff values. If her data in fact consist of some factors that are mostly ergodic, some factors that are strongly predominant between and some that are strongly predominant within participants, she would expect a scree plot that is first close to one then skips sharply to zero, and then skips to −1 again as the between-participant factors are reached. To contrast this with a smooth case, we simulated three situations that are homogeneously ergodic or non-ergodic (that is, where we expect the scree plot to be smooth).

In this simulation, we generated two covariance matrices for K=9 variables independently from Wishart distributions with 2N degrees of freedom. We then created data in three conditions, either by using both independent matrices (resulting in a non-ergodic case), by using only one of the matrices (either the between or within participant matrix, resulting in a fully ergodic case), or by mixing the within matrix with 50% from the between matrix and 50% from the independent matrix (an intermediately ergodic case).

Figure 3 shows the scree plot for the three cases. The fully ergodic line is always zero, and the random line ranges from 0.8 to −0.8 fairly linearly. Note that the values are not expected to be 1.0 or −1.0 since both covariance matrices were sampled independently and are therefore expected to show some more and some less ergodic factors. The plot for the intermediate condition shows a slight curvature, being steeper at both ends of the scree plot. All three curves intersect with the zero line almost perfectly at the fifth variable.

The position of an empirical scree plot curve from real data relative to the theoretical curves in this diagram indicates the ergodicity situation found in the data. If the empirical curve follows one of the three trajectories, we have a fairly smooth transition from between to within factors. The steeper the curve is, the less ergodic the data is overall. If the curvature is stronger, that is, more values lie in the corridor between the two cutoff values (for example, between −0.1 and 0.1 for the cutoffs suggested above), the data can more clearly be separated into ergodic and non-ergodic parts.

## 5. Application to COGITO Data

In this section, we apply ESA to a real cognitive dataset. The data is a subset of the COGITO dataset collected by [27], in which 101 adults aged 20 to 31 years completed twelve cognitive tasks (only nine are considered here) on a little more than 100 daily occasions. [28] demonstrated the presence of reliable day-to-day fluctuations in cognitive performance within individuals.

Participants performed the tasks individually in a lab room. They could come to the lab on up to six days each week until they completed 100 sessions. On average, this took them 197 days in total. Before and after the training period, participants did the same tasks as pre- and post-tests, together with a number of other measurements not included in this analysis. We used these pre-tests for the between-person covariance matrix in the later analysis. The internal review board of the Max Planck Institute for Human Development, Berlin, approved the study.

The nine cognitive variables consisted of three blocks related to perceptual speed, episodic memory, and working memory, with three tasks in each block. In each of these three blocks, one task was a numeric task, one a verbal task, and one a figural task. We omit here some details about the tasks that are not central to this analysis (for more details, see [29]).

The total size of the dataset was N=100 participants with over T=100 time points on K=9 variables. The classical literature suggests a between-person factor of general intelligence and weaker factors for the three cognitive domains. [29] confirm this between persons but find that this is not reflected in the within-person structures (cf. [7]). In fact, they find that the within-person structures are, at least for some participants, substantially different from the between-person structure. To analyze this question further and to move it to a quantitative evaluation of ergodicity, we use the ESA on this dataset.

In preparation, we de-trended the data using a Gauss filter with a standard deviation of three sessions to remove long-term training effects. We then performed the analysis both on the raw data and the de-trended data. We computed an average between and within covariance matrix from the data and used this as input to the ESA.

Figure 4 shows the same plot as Figure 3 with the two scree lines from the COGITO data added. As can be seen, the COGITO data corresponds to the intermediate situation; that is, roughly half of the variance in the COGITO data is non-ergodic on top of what one would expect randomly. Both the de-trended and the raw data show reasonably similar results. For the de-trended data, two factors emerged that were clearly dominant in the between variance, three that were mostly ergodic (with ergodicity values from 0.004 to −0.081), and four that were strongly dominant within persons. The full factors are shown in Table 3.

The strongest factor with dominant variance between persons is the general *g* factor, representing the positive manifold of cognitive abilities. We can see this from the uniform positive loadings. Note that the factor is the furthest away from ergodicity among the between factors. That is, its variance comes predominantly from the between structure but is much less present in the within structure, although the value of $erg_{1}=0.314$ still suggests that there is some ergodic part even in this factor. The strongest factor within participants ($erg_{9}=-0.542$), on the contrary, shows strong loadings both in the positive and negative direction. This factor seems to mostly contrast the numeric from the verbal (and to a lower degree the figural) aspect in the processing speed domain. That is, participants who on one day performed stronger on the numeric processing speed tasks seem to perform weaker on the other two processing speed tasks. However, this dependency is not seen between participants. The second strongest within factor that is also dominated by within variance ($erg_{8}=-0.460$) seems to work similarly in the working memory domain, again contrasting numerical to verbal parts, although figural parts seem less involved here. The factor that represents the most ergodic one-dimensional subspace among these variables ($erg_{3}=0.004$) contrasts the figural from the numeric and verbal aspect in episodic memory, and to a much weaker degree also in processing speed. The total subspace spanned by the factors 3 to 5 spans a three-dimensional subspace of the variables with a reasonable ergodic structure. Note that the ergodicity values are only relative statements; for example, the value $erg_{9}=-0.542$ just shows that this factor is much more present within a person than between persons, but no statement about its variance is made.

## 6. Discussion

We introduced ergodic subspace analysis (ESA), a method to factor the space of variables measured between and within persons into an ergodic subspace and two spaces predominantly within or predominantly between persons. For all factors, an ergodicity value is computed that lies between −1 (for factors dominant within persons) to 1 (for factors dominant between persons). Ergodicity values close to zero indicate ergodic factors. In particular, the method allows us to determine how much ergodicity is present in a dataset overall (via the slope of the scree function), whether the overall structure is more a smooth transition from between to ergodic to within factor or rather a stepwise function, and in which dimensions variance sources within persons or variance sources between persons are stronger.

ESA is comparable to multilevel confirmatory or exploratory factor analysis, including the more general multilevel structural equation modeling (SEM) framework (cf. [21]; [22]) in the sense that all approaches allow the estimation of factors at different levels. ESA shares the idea of decomposing the total covariance matrix into a between-person and a within-person part with several prominent approaches in multilevel factor analysis, such as the pseudo-balanced approach by [19], [20], the weighted least square (WLS)-based approach for categorical variables ([2]), and the two-step approach of estimating the between-person and within-person covariance matrix via a multivariate multilevel model followed by a full SEM analysis ([9]; see also [10]). For an overview and comparison of approaches with full maximum likelihood (ML) estimation (cf. [17]) and other variants, see [13]; [14]. Although it is related, the purpose of ESA is rather different and more specific: Instead of modeling factor structures at different levels, ESA identifies subspaces according to their degree of ergodicity. The approach is primarily data-driven and offers a new way to quantify the degree of ergodicity. While we believe the approach is useful by itself, in case of panel data it could also serve as an initial step towards a more elaborate analysis of multilevel structures using any of the approaches outlined above.

The method is an adaptation of the common Spatial patterns (CSP) method; it can be implemented in any programming language where PCA functionality is available (e.g., in Python, MATLAB, or R). The application of dimension reduction in the context of ergodicity has not been done before, to the best of our knowledge. ESA can be used with different methods to obtain between and within covariance matrices from a single time series and one cross-sectional dataset in the minimal case to a full N×T data matrix for maximal information.

We demonstrated the method on five examples, two computational and three Monte Carlo simulations. All five cases show that when the ground truth is known, that is, when data is generated explicitly from between, within, or ergodic factors, ESA can retrieve these factors. The standard error of the ergodicity values is not negligible for smaller sample sizes but reduces quickly with sample size, while the lengths of the time series are of less importance. When overall ergodicity is smaller, the precision of the ergodicity value is higher, but at the same time it is harder to identify the factors, down to an average cosine of roughly one half for the angle between the true factor and the recreated factor if the factor is 95% ergodic.

ESA allows the quantification of the concept of ergodicity, moving it from a categorical classification of “ergodic” and “non-ergodic”, which are both almost non-existent in their pure form in nature. In particular, we showed for rather general data from the cognitive domain ([27]) that cognitive data is roughly as ergodic as a 50:50 artificial mixture. That is, to the degree that COGITO can be generalized to cognitive functioning in general, roughly half of the variance in the between and within structure of cognitive abilities can also be found in the respective other, in addition to the degree of similarity expected from independently drawn covariance matrices. It is also interesting to observe that the general ability factor of cognition is the factor that is most strongly limited to the between-person domain and shared within persons to the smallest degree; this is in line with the results found by [29]. The within factors found in the COGITO data, in particular the strongest within factor, can be seen as contrast factors between numerical and verbal (and to some degree, figural) tasks in terms of processing speed: If participants are good at performing the numerical task on one day, they will do worse on the verbal task and vice versa. This directly corresponds to our initial examples, both with the typewriter and the sports example. This may be interpreted in a way that, in an analogy to the examples, a general resource available to the participants is either more exhaustively spend on the numerical or the verbal task.

Note that the result of the ESA is a relative statement, relating the strengths of a factor between and within persons. So although it may be unlikely in practice, it could be that the factor that explains the most variance within persons nevertheless appears as a predominantly between-person factor in the ESA because the factor explains an even higher portion of the variance between persons. In the same vein, if an ergodic factor is found in the ESA, that indicates that the factor has roughly the same strength in the between and the within domain; no statement can be made about the variance explained by this factor. It could equally well be a factor that is virtually the only variance source for the variables in both domains, or it could be a factor of negligible explanatory power in both. However, it is of course possible to compute the explained variance in each domain and include this in the interpretation.

Although ESA can be used with a single time series and one cross-sectional dataset, these can still be difficult to obtain and in fact can be misleading if participants, or less frequently time points, do not show identical marginal distributions. As is often the case (cf. [16]), a full panel design is preferable to obtain generalizable results. Albeit that such a full dataset is always a resource-costly project, we showed that the number of time points is less critical than the number of participants (with T=25 usually showing almost no less accuracy than longer time series), suggesting that even for behavioral data, such projects could be feasible.

If the marginal distributions differ between participants, as is the case in the COGITO dataset, the researcher needs to decide on the within-person matrix the between-person matrix should be compared to. For some research questions, it might be interesting to investigate ergodicity with respect to every single participant (e.g., [30]), for example, to cluster them into groups with a similar strength or structure of ergodic factors. For other research situations, it may be useful to use a central within-person covariance matrix (e.g., the average of all within-person matrices) even if the marginal distributions differ, to identify factors that are ergodic between the time points and the participants as a whole group.

ESA only operates on the first two moments of the respective distributions; even though it can also be used on non-normal data, higher-order moments will not be reflected in the factorization.

ESA introduces a quantitative measure of ergodicity and allows users to investigate where in their data space between, within, or ergodic factors can be expected. As mentioned earlier, ESA can be seen as an exploratory sibling of conditional equivalence, which is also aimed at finding ergodic subspaces, but is defined by controlling for pre-postulated covariates. Together, both approaches allow a new theoretical perspective on the question of the degree to which results gathered between participants can be applied to understanding a single person’s past or to predict her future.   

## Figures and Tables

**Figure 1 jintelligence-08-00003-f001:**
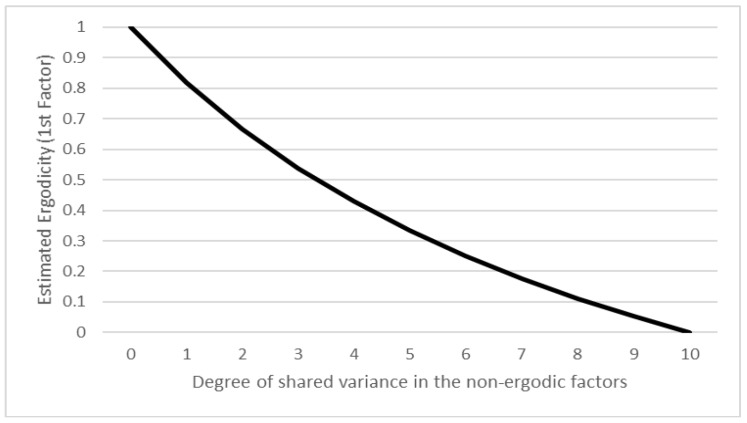
Expectation of the first estimated ergodicity value for different *x* values in the simulation. The ergodicity values follow ${\left(1-x\right)}^{2}$.

**Figure 2 jintelligence-08-00003-f002:**
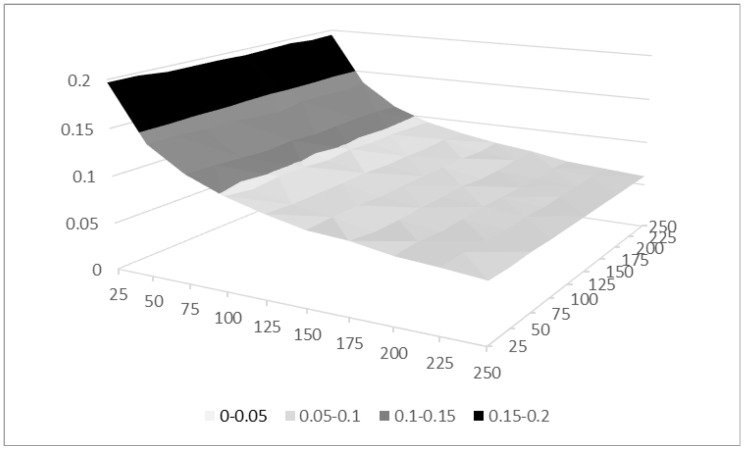
The expectation of the first estimated ergodicity value for different sample sizes *N* and time point *T*.

**Figure 3 jintelligence-08-00003-f003:**
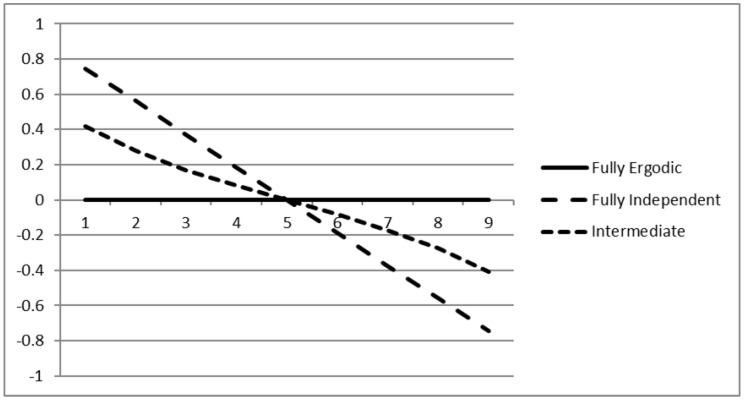
Scree plot for three situations of ergodicity: fully ergodic (independently sampled covariance matrices), fully non-ergodic (identical covariance matrices), and an intermediate case with a 50% mixture.

**Figure 4 jintelligence-08-00003-f004:**
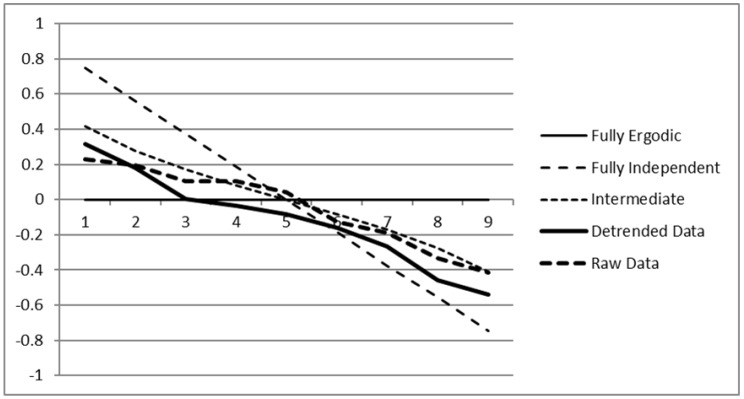
Scree plot for three artificial situations of ergodicity, fully ergodic (independently sampled covariance matrices), fully non-ergodic (identical covariance matrices), and an intermediate case with a 50% mixture, together with the ergodicity values of the nine cognitive tasks from the COGITO study (raw and de-trended).

**Table 1 jintelligence-08-00003-t001:** Reconstruction precision of the factor with dominant variance between participants and its corresponding ergodicity value. The value *x* gives the degree to which the data is ergodic, with x=0 indicating strong differences between and within participants and x=10 indicating a perfectly ergodic situation. The angle α is the angle between the true first factor and the reconstructed factor; a cosine of one means perfect reconstruction, a cosine of 0 means orthogonal vectors.

*x*	cos(α)	${\mathrm{erg}}_{1}$	stdv(${\mathrm{erg}}_{1}$)
0	1	1	0.144
1	0.999	0.823	0.129
2	0.996	0.670	0.119
3	0.988	0.542	0.111
4	0.972	0.432	0.103
5	0.938	0.337	0.095
6	0.875	0.253	0.090
7	0.764	0.182	0.084
8	0.594	0.115	0.080
9	0.421	0.056	0.075
10	0.277	0.005	0.072

**Table 2 jintelligence-08-00003-t002:** Standard error of the ergodicity value of the factor with dominant variance between participants for different sample sizes and time series lengths. High values indicate uncertain measurements, while lower values indicate a more precise measurement of the degree of ergodicity in the data.

	*N*									
T	25	50	75	100	125	150	175	200	225	250
25	0.197	0.139	0.114	0.098	0.087	0.078	0.075	0.068	0.065	0.061
50	0.197	0.139	0.113	0.094	0.085	0.079	0.071	0.069	0.066	0.060
75	0.194	0.136	0.111	0.097	0.085	0.079	0.071	0.066	0.064	0.061
100	0.194	0.137	0.109	0.097	0.085	0.077	0.072	0.067	0.063	0.060
125	0.194	0.133	0.111	0.094	0.086	0.078	0.072	0.068	0.064	0.060
150	0.193	0.131	0.109	0.097	0.084	0.078	0.073	0.067	0.062	0.060
175	0.194	0.135	0.110	0.095	0.085	0.078	0.070	0.067	0.063	0.060
200	0.195	0.135	0.111	0.096	0.084	0.078	0.070	0.066	0.063	0.060
225	0.193	0.133	0.111	0.095	0.086	0.076	0.072	0.068	0.064	0.060
250	0.194	0.136	0.109	0.094	0.085	0.078	0.073	0.066	0.063	0.060

**Table 3 jintelligence-08-00003-t003:** The ergodicity values and the corresponding nine factors of the ergodic subspace analysis (ESA) on the de-trended cognitive data from the COGITO study.

erg	Processing Speed			Episodic Memory			Working Memory		
	Numerical	Verbal	Figural	Numerical	Verbal	Figural	Numerical	Verbal	Figural
0.314	0.200	0.155	0.130	0.144	0.191	0.141	0.113	0.166	0.214
0.181	0.125	0.201	0.165	0.040	0.009	0.194	−0.370	−0.334	−0.037
0.004	0.246	0.076	0.206	−0.331	−0.237	−0.379	0.140	0.092	0.066
−0.038	−0.041	−0.201	0.120	−0.343	−0.234	0.405	−0.065	−0.076	0.398
−0.081	0.032	0.170	−0.229	0.393	−0.551	−0.019	−0.004	−0.009	0.163
−0.163	0.177	0.400	−0.619	−0.334	0.095	0.096	−0.118	0.120	−0.004
−0.264	0.119	0.016	0.062	−0.088	−0.245	0.421	0.282	0.116	−0.58
−0.460	0.115	0.043	−0.169	−0.017	0.083	−0.035	0.591	−0.671	0.078
−0.542	0.785	−0.652	−0.251	0.108	0.039	0.009	−0.119	−0.020	−0.013

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
