# Peer review of "Ergodic Subspace Analysis"

_jintelligence, 2020, doi:10.3390/jintelligence8010003_

Round 1
Reviewer 1 Report
Review of "Ergodic Subspace Analysis"
=====================================
The paper at hand introduces a technique dubbed "Ergodic Subspace Analysis" (ESA) which is an adaption of a spatial technique (CPS). It is intended to find variables that behave similar on a within-person and between-person level, especially for data from panel studies. While the paper shys aways from a systematic presentation from the related literature and could be more precise in its presentation of the material, the method is easy enough to implement and interpret to be useful in the intended setting and merits publication.
Major Remarks
=============
1. I feel that at least a brief literature review of techniques for multilevel data with both within and between person factors is indicated. Especially, multilevel exploratory factor analysis (MFA; e.g. Goldstein et al., 2005, and many papers from Reise) seems to be well-known and useful enough to be a reference method in the current context. Similarly, (exploratory) structural equation models exist for the same purpose and I am almost certain both authors are intimately familiar with these techniques.
2. That said, the example would profit from a direct comparison with other methods, say with MFA. This would convinve readers that there is an added value of applying ESA.
3. The exposition of ESA would profit from being more precise (w.r.t. to some definitions, see below, and w.r.t. to the generalized eigenvalue problem that ESA solves). I guess you are writing for a mixed audience, and while part of it could maybe live with the somewhat vague introduction ("structures"), precise definitions would help others.
4. Maybe I am missing something, but as ESA is essentially focusing on the PCs of the pooled covariance matrix Sigma, would it not be natural to rotate the BP and WP factors, say to obtain a simple structure of BP factors and (subsequently resp. parallely but not jointly) of the WP factors. This could aid to understand the three subspaces of (almost) ergodic factors. Another strategy would be to try and find an orthogonal rotation of the (almost) ergodic factors to understand what is going on.
Minor Remarks
=============
- Abstract: Please formally introduce the ESA acronym
- Abstract: "Eigenvalues" -> "eigenvalues"
- Abstract: "The difference of the Eigenvalues allows to separate the rotated
dimensions ....": I believe I missed the part in the ms. where dimensions are rotated (see above).
- Abstract/Introduction: The abstract (implicitly) defines ergodicity via the equality of covariances, while the first paragraph of the introduction talks about correlations. Please be more consistent.
- Introduction: "... this is not true for specific persons over time, where considerable divergences from the g factor model can be observed": It could help some readers to mention how individual divergences from a g factor model are assessed. I assume that won't be thinking of 100 days but longitudinal work spanning more time.
- After Eq. 5: "weak stationarity": it would be good to clarify what is meant by "weak ergodicity" in the context of this paper, too
- After Eq. 7: "again for a fixed covariance matrix Σ" I guess a bit more precise would be "for the same fixed covariance matrix Σ"
- p.7: "Hence, no covariance matrix for a specific person at a specific time point can be estimated, and since every exception strictly speaking destroys ergodicity, the strict version of ergodicity cannot be tested." I have two remarks: (i) Please clarify what exceptions you are referring to here. (ii) I think the wording is a bit too strong. As you correctly mention further up, necessary consequences of strict ergodicity can be tested. Maybe change "tested" to "confirmed"?
- p.9: The paragraph beginning with "Voelkle et al. (2014) suggest ..." has only one sentence. Consider to merge with the previous paragraph. Similarly on p. 10 for "Once we obtained ..."
- Whitening Step: It would help some readers to mention the Cholesky decomposition here.
- Analogy to CSP: I believe that similar to CSP, there is a generalized eigenvalue problem that the steps solve. Please make that explicit.
- Some of the Figures appear to be low quality. Please submit vector graphics if possible.
- References: Hamaker (2012) has some redundant dots above a "w" and a ":"
- References: Pearson (2012) is really Pearson (1901)
References
==========
Goldstein H, Browne W. Multilevel factor analysis models for continuous and discrete data. In: Maydeu-Olivares A, McArdle JJ, editors. Contemporary psychometrics: A festschrift for Roderick P. McDonald. Mahwah, MJ: Erlbaum; 2005. pp. 453–475.
Reviewer 2 Report
See attached file

Reviewer 3 Report
The concept of ergodicity is introduced with helpful examples like the contrast between typing speed and accuracy or between research and dissemination.
I don't understand the definition of ergodicity on page 6. Covariance matrices $\Sigma$ are indexed by person and by time. We might form a covariance matrix over K variables in columns and persons in rows. If $\Sigma$ is both person and time specific, K variables are in the columns then what are the rows? To assess variation, we need at least 2 measurements of each variable, yes? You seem to address this on page 7? No, it is a different issue. Ah, your manual computation example on page 13 is clarifying things. You actually have two time scales: within typewriter sheet and between typewriter sheet. So you should definitely clarify that time is further subdivided.
Overall, I think this is an interesting paper with a mostly well presented mathematical technique.
Round 2
Reviewer 1 Report
Thank you for the revised version which is now more accessible. My comments have all been adequately addressed, if they did not lead to major changes in the manuscript.
Reviewer 3 Report
Looks good